# The Steric Effect in Preparations of Vanadium(II)/(III) Dinitrogen Complexes of Triamidoamine Ligands Bearing Bulky Substituents

**DOI:** 10.3390/molecules27185864

**Published:** 2022-09-09

**Authors:** Yoshiaki Kokubo, Itsuki Igarashi, Kenichi Nakao, Wataru Hachiya, Shinichi Kugimiya, Tomohiro Ozawa, Hideki Masuda, Yuji Kajita

**Affiliations:** 1Faculty of Engineering, Aichi Institute of Technology, 1247 Yachigusa, Yakusa-cho, Toyota 470-0392, Japan; 2Graduate School of Engineering, Nagoya Institute of Technology, Gokiso-cho, Showa-ku, Nagoya 466-8555, Japan

**Keywords:** dinitrogen complex, vanadium, bulky substituents, dinitrogen activation, steric effect, triamidoamine ligand

## Abstract

The reactions of newly designed lithiated triamidoamines Li_3_**L^R^** (**R** = iPr, Pen, and Cy_2_) with VCl_3_(THF)_3_ under N_2_ yielded dinitrogen–divanadium complexes with a μ-N_2_ between vanadium atoms [{V(**L^R^**)}_2_(μ-N_2_)] (**R** = iPr (**1**) and Pen (**2**)) for the former two, while not dinitrogen–divanadium complexes but a mononuclear vanadium complex with a vacant site, [V(**L^Cy2^**)] (**R** = Cy_2_ (**3**)), were obtained for the third ligand. The V–N_N2_ and N–N distances were 1.7655(18) and 1.219(4) Å for **1** and 1.7935(14) and 1.226(3) Å for **2**, respectively. The ν(^14^N–^14^N) stretching vibrations of **1** and **2**, as measured using resonance Raman spectroscopy, were detected at 1436 and 1412 cm^–1^, respectively. Complex **3** reacted with potassium metal in the presence of 18-crown-6-ether under N_2_ to give a hetero-dinuclear vanadium complex with μ-N_2_ between vanadium and potassium, [VK(**L^Cy2^**)(μ-N_2_)(18-crown-6)] (**4**). The N–N distance and ν(^14^N–^14^N) stretching for **4** were 1.152(3) Å and 1818 cm^−1^, respectively, suggesting that **4** is more activated than complexes **1** and **2**. The complexes **1**, **2**, **3**, and **4** reacted with HOTf and K[C_10_H_8_] to give NH_3_ and N_2_H_4_. The yields of NH_3_ and N_2_H_4_ (per V atom) were 47 and 11% for **1**, 38 and 16% for **2**, 77 and 7% for **3**, and 80 and 5% for **4**, respectively, and **3** and **4**, which have a ligand **L^Cy2^**, showed higher reactivity than **1** and **2**.

## 1. Introduction

Dinitrogen activation using vanadium ions has been intensely investigated by bioinorganic chemists and coordination chemists in order to understand the role of vanadium ions as an important factor in vanadium nitrogenase enzymes [1,2,3,4,5,6,7,8,9]. Most of the dinitrogen–vanadium complexes studied are dinuclear complexes with a μ-N_2_ ligand in the end-on mode [10]. On the other hand, mononuclear vanadium–dinitrogen complexes are scarce [11,12,13,14]. It has also been previously reported that some dinitrogen–vanadium complexes produce ammonia and hydrazine [15,16,17,18,19,20,21,22]. Recently, dinitrogen–vanadium complexes, which are supported with anionic pyrrole-based PNP-type pincer and aryloxy ligands, have been successfully used for catalytic dinitrogen reduction by a group of Nishibayashi [20].

Triamidoamine (tris(2-amidoethyl)amine) ligands are very useful ligands that bind as multidentate ligands when forming metal complexes, creating binding sites for external ligands on the axis. Therefore, many complexes with the ligands have been investigated [14,21,22,23,24,25,26,27,28,29,30,31,32,33,34,35,36,37,38,39,40,41,42,43]. Schrock and co-workers reported the molybdenum complex with the triamidoamine ligand with a very bulky substituent group, HIPT (3,5-(2,4,6-iPr_3_C_6_H_2_)_2_C_6_H_3_) [37]. This molybdenum complex is the first example of catalytic ammonia production [38]. In this complex, the HIPT group functioned to inhibit dimerization of the molybdenum center and provide a pocket for binding an external ligand, such as N_2_, HNN, H_2_NN, NH, NH_3_, N, NO, THF, CO, S, Me_3_SiN, and so on [14,37,39,40]. Additionally, they also argued that the N_2_ ligand is protonated in the distal pathway by steric hindrance of the HIPT group [39]. On the other hand, dinuclear dinitrogen–molybdenum complexes with triamidoamine ligands have also been studied, but the substituents of these triamidoamine ligands are smaller than HIPT, such as TMS, aryl, and alkyl groups [21,22,41,42,43,44], and those with larger substituents have not been studied.

We have previously reported the syntheses and crystal structures of the dinitrogen–divanadium complexes bearing a triamidoamine ligand with a secondary C atom on the terminal N atom [{V(**L^R^**)}_2_(*μ*-N_2_)] (R = iBu, EtBu, iPr_2_Bn, Bn, MeBn) and studied the conversion of the bridging N_2_ ligand to ammonia using these complexes in the presence of proton sources (HOTf, [LutH](OTf)) and reductants (M^+^[C_10_H_8_]^-^ M = K or Na) [21,22]. Furthermore, the crystal structure and protonation reaction of its Na^+^ adduct ([Na{V(**L^iBu^**)}_2_(*μ*-N_2_)]) were also studied (Figure 1) [22]. By introducing a secondary carbon atom on the terminal nitrogen atom of the trimidoamine ligand, these dinitrogen complexes can easily form dimer structures because the steric hindrance around the vanadium ion is smaller than those modified with bulky silyl or aryl groups. Therefore, we considered it would be possible to systematically investigate the structure and reactivity of mononuclear or dinuclear dinitrogen complexes using a series of triamidoamine ligands with different steric hindrances. The space-filling models of previously reported divanadium–dinitrogen complexes [{V(**L^R^**)}_2_(*μ*-N_2_)] (R = iBu and MeBn) are shown in Figure 1. In these complexes, there is no space around the secondary carbon atom (red), suggesting that a tertiary carbon atom was introduced on the terminal N atom to form a mononuclear vanadium–dinitrogen complex. If mononuclear and dinuclear complexes can be synthesized using triamidoamine ligands with the same backbone, meaningful comparisons can be made regarding their structures and reactivities in dinitrogen activation.

In this study, three vanadium complexes with a series of triamidoamine ligands bearing bulky substituents were synthesized under N_2_ atmosphere, and the characterizations, crystal structures, and protonation reactivities of the obtained complexes were investigated, compared, and discussed with those previously reported.

## 2. Results and Discussion

### 2.1. Syntheses of Ligands and Their Vanadium Complexes ***1***, ***2***, and ***3***

Three types of tren derivatives **H_3_L^R^** (**R** = iPr (tris(2-isopropylaminoethyl)amine, **H_3_L^iPr^**), Pen (tris(2-(3-pentylamino)ethyl)amine, **H_3_L^Pen^**), and Cy_2_ (tris(2-dicyclohexylmethylaminoethyl)amine, **H_3_L^Cy2^**)) were prepared using previously reported methods [21,22]. **H_3_L^iPr^** and **H_3_L^Pen^** were obtained as light-yellow oil and **H_3_L^Cy2^** as colorless crystals, which were characterized using ^1^H NMR, ^13^C NMR, and IR spectroscopic methods. The **H_3_L^R^** was deprotonated and used as a triamidoamine ligand for the synthesis of dinitrogen complexes.

The reactions of lithiated triamidoamines **Li_3_L^R^** (**R** = iPr, Pen, and Cy_2_) with VCl_3_(THF)_3_ at room temperature under N_2_ produced dinitrogen–divanadium complexes [{V(**L^R^**)}_2_(*μ*-N_2_)] (R = iPr (**1**) and Pen (**2**)) for the former two. On the other hand, for the third ligand, a mononuclear vanadium complex [V(**L^Cy2^**)] (**3**) was obtained instead of a dinitrogen–divanadium complex (Figure 1). When all complex solutions were left at room temperature for several days, single crystals of complexes **1** and **3** were obtained as dark green crystals, and that of complex **2** was observed as dark purple crystals. These complexes were stable at low temperature under N_2_ atmosphere but decomposed under air atmosphere.

### 2.2. Crystal Structures of ***1*** and ***2***


The crystal structures of **1** and **2** are shown in Figure 2, and the crystal parameters are listed together in Table 1 and Appendix A. Complexes **1** and **2** were expected to be mononuclear vanadium complexes because of their bulky substituents, but they turned out to be dinuclear vanadium complexes with bridging dinitrogen in the end-on mode. The coordination geometries around the vanadium centers in **1** and **2** are a nearly undistorted triangular bipyramid (*τ* = 1.0 and 1.0, respectively) (Figure 2), where 1 is a perfect trigonal bipyramidal geometry, 0 is a perfect square pyramidal geometry [45]. The N–N bond lengths for **1** and **2** are 1.219(4) and 1.226(3) Å, respectively, and that of complex **2** are slightly longer than those of **1** and the previously reported divanadium–dinitrogen complexes [21,22]. The bond lengths around the vanadium center in **2** (V–N_N2_ (1.7935(14) Å), V–N_amido_ (1.9276(16), 1.9284(16), 1.9234(16) Å), and V–N_amine_ (2.1854(16) Å)) are also more elongated than those of **1** (V–N_N2_ (1.7647(18) Å), V–N_amido_ (1.896(2), 1.9123(14), 1.9123(14) Å), and V–N_amine_ (2.173(2) Å)). This may also be due to the greater steric repulsion between the alkyl substituents on the N atoms of the triamidoamine ligand in **2** than in **1**. Comparing the space-filling model of **2** with that of **1**, it is obvious that the pentyl group in **2** surrounds the vanadium center more than the isopropyl group in **1** (Figure 3). Thus, it appears that all bond lengths are extended to maintain the dimeric structure, overcoming the pull away to the mononuclear vanadium complex. In fact, the V•••V distances and the distances of the vanadium ion from the plane decided by three N_amido_ atoms are 4.7482(8) and 0.3055(12) Å for **1** and 4.8128(7) and 0.3495(10) Å for **2**, respectively, making them longer for complex **2** than for complex **1**. This finding appears to be due to the strong attraction of vanadium ions in complex **2** to the μ-N_2_ ligand.

The μ-N_2_ ligands in **1** and **2** are stabilized by hydrogen bonding interactions between the N_2_ ligand and the methine hydrogen atoms on the N atoms (CH•••N_N2_ = av. 2.488 Å for **1**, av. 2.611 Å for **2**). This interaction may also contribute to the formation of dimer structures.

### 2.3. Crystal Structure of ***3***

The crystal structure of complex **3** is shown in Figure 4, and the crystal parameters are listed in Table 2 and Appendix A, respectively. Complex **3**, unlike **1** and **2**, was a mononuclear vanadium complex with no dinitrogen coordination. The geometry around the vanadium center is a trigonal pyramid, with a vacancy on the opposite side of N_amine_ (N5) in [**L^Cy2^**]^3-^. The V1–N3 (1.9433(12) Å) and V1–N4 bonds (1.9593(12) Å) are longer than the V1–N2 bond (1.9281(13) Å) because the steric repulsions between the cyclohexane rings of the dicyclohexylmethyl group on the N2 atom and those on the N3 and N4 atoms are weakened by the overhang of those on the N2 atom attached to the V center. The averaged V–N_amido_ bond length in **3** (1.9436 Å) is longer than that of the related triamidoamine–vanadium complex, [V(*t*BuMe_2_SiN)_3_N] (V–N_amido_ = 1.930(av.) Å) [26]. This fact indicates that the dicyclohexylmethyl group of **3** leads greater steric repulsion than the *t*BuMe_2_Si group. 

### 2.4. Synthesis and Crystal Structure of ***4***

The crystal structure of **3** shows that there is an open site on the vanadium ion. Therefore, we attempted to synthesize the N_2_ adduct by reacting **3** with potassium metal under N_2_ in the presence of 18-crown-6-ether (Figure 2). Fortunately, a single crystal of the N_2_ adduct (**4**) was obtained as green crystals using recrystallization from THF/hexane at −35 °C. Complex **4** gradually decomposed at room temperature even under inert gas (N_2_ or Ar).

The crystal structure of **4** is shown in Figure 5, and the crystal parameters are listed in Table 2 and Appendix A, respectively. Complex **4** had a bridging dinitrogen ligand between the vanadium(II) ion and potassium ion in the end-on mode, the coordination geometry around the vanadium center was trigonal bipyramidal, and the THF molecule coordinated to the potassium ion from the opposite side of the N_2_ ligand. The N–N and V–N_N2_ bond lengths of **4** are 1.152 (3) and 1.853 (3) Å, respectively. The average of three V–N_amido_ bond lengths was found to be 1.960 Å, which is more elongated than those of **3** (1.9436 (av.) Å). This is thought to be due to the increased steric repulsion between the dicyclohexylmethyl groups as a result of the increased ionic radius of the vanadium ion due to the reduction in V(III) to V(II) and the incorporation of the N_2_ ligand in the axial position. The N_2_ ligand of **4** was also stabilized by hydrogen bonding interactions between the methine proton and the N3 atom of dinitrogen (CH•••N3(N_2_) = av. 2.596 Å). The distance between the mean plane decided by three N_amido_ atoms and the vanadium ion (V1 atom) was 0.2888(14) Å, which is smaller than those of **1** and **2**. The space-filling model of **4** is shown in Figure 5 (right). It is clear from this figure that the dicyclohexylmethyl group surrounds not only the α-nitrogen but also the β-nitrogen, suggesting that the dinitrogen–vanadium complex with [**L^Cy2^**]^3-^ ligand is too large to form a dimer structure.

### 2.5. Raman and Infrared Spectra of ***1***, ***2***, and ***4***

The ν(^14^N–^14^N) stretching vibrations of **1** and **2** were detected at 1436 and 1412 cm^–1^ using resonance Raman spectroscopic measurements, respectively (Appendix A). ^15^N-labeled **1** (**1′**) and **2** (**2′**) were both split by a Fermi doublet to show peaks at 1399, 1337 cm^–1^ and 1380, 1335 cm^–1^, respectively. The ν(V–^14^N) stretching vibrations of **1** and **2** were observed at 796 and 728 cm^–1^ from the IR spectra, respectively (Appendix A). The ν(^14^N–^14^N) and ν(V–^14^N) values of **1** are larger than those of **2**, which are in good agreement with the N–N and V–N_N2_ bond lengths trends for **1** and **2**. However, these ν(^14^N–^14^N) values are larger than those of previously reported divanadium–dinitrogen complexes (1394–1402 cm^−1^), even though **1** and **2** have longer N–N bonds than those of previously reported divanadium–dinitrogen complexes (1.200 (5)–1.226 (3) Å) [21,22]. Such inversions in bond lengths and vibrational spectra are sometimes observed in the activation chemistry of dinitrogen with transition metals [46].

In the IR spectral measurements, the *ν*(^14^N–^14^N) peaks for **4** were observed as two bands at 1830 and 1818 cm^–1^, which were shifted to 1768 and 1759 cm^–1^ when ^15^N_2_ was used in the place of ^14^N_2_ (Appendix A). The IR bands at 1830 and 1768 cm^−1^ were assigned as overtones of the 910 and 885 cm^−1^, respectively. When ^15^N-labeled **4** (**4′**) was dissolved in THF at room temperature and recrystallized under ^14^N_2_, the *ν*(^14^N–^14^N) stretching vibration was observed. This means that the N_2_ ligand of **4** is easily exchanged in THF because of the weak V–N_N2_ bond. 

### 2.6. ^1^H-, ^15^N-, and ^51^V-NMR Spectra 

^1^H NMR spectra of **1** and **2** exhibited sharp peaks in the diamagnetic region, as shown in Appendix A, respectively. The methine peak of **2** (4.77 ppm) was observed in a higher magnetic field region than that of **1** (5.27 ppm). ^15^N NMR spectra of **1** and **2** with ^15^N-labeled N_2_ were detected at 25.2 and 41.6 ppm (Appendix A), and ^51^V NMR spectra of **1** and **2** were observed at −211 and −47.6 ppm, respectively (Appendix A). The peaks of **2** were both observed in a lower magnetic field region than those in **1** and our previously reported divanadium–dinitrogen complexes (^15^N NMR: 25.2–33.4 ppm, ^51^V NMR: −240.2−143.8 ppm) [21,22]. These findings indicate that the electron densities on the N atoms of dinitrogen and the V atom in **2** are lower than those of complex **1** and the previously reported divanadium–dinitrogen complexes because the electron donation from N_amido_ atoms to the V atom in **2** was smaller than those complexes. These facts correspond well with the result that the V–N_amide_ bond length is the longest among the dinuclear vanadium–dinitrogen complexes with triamidoamine ligands reported so far [21,22], due to the large steric repulsion between the pentyl groups. On the other hand, the ^1^H NMR spectrum of **3** gave a broadened paramagnetic peak at 915 ppm in the lower magnetic field region (Appendix A) and that of **4** showed broad peaks at 10.2, 3.28, 0.32, 0.08, −0.28, −0.75, −0.95, −1.63, −15.2, and −30.5 ppm in the diamagnetic to higher magnetic field region (Appendix A). The spectrum of **4** includes not only **4** but also solvents (*n*-hexane, THF) and a free ligand, **H_3_L^Cy2^** (Appendix A). It is thought that the free ligand was probably produced by the decomposition of **4** due to its low thermal stability at room temperature. Unfortunately, we were unable to characterize the products containing vanadium(II) ions produced in the decomposition of **4**. The effective magnetic moment (μ_eff_) of **3** was 2.73*μ*_B_ at 298 K as determined by Evans’s NMR solution method [47,48], which indicates that the spin state of **3** is *S* = 1. Similarly, μ_eff_ of **4** was estimated to be 1.76*μ*_B_ at 298 K, indicating that the spin state of **4** is S = 1/2. These spin states correspond well with the structural findings that complex **3** is a mononuclear vanadium(III) complex with a vacant site, [V(**L^Cy2^**)], and complex **4** is a hetero-dinuclear vanadium(II) complex with *μ*-N_2_ between vanadium and potassium, [VK(**L^Cy2^**)(*μ*-N_2_)(18-crown-6)].

### 2.7. Protonation of ***1***–***4*** in the Presence of Reductants

Protonation of **1**, **2**, **3**, and **4** was carried out with a reductant (M[C_10_H_8_] (M = Na^+^ or K^+^)) and an acid (HOTf) in THF at −78 °C. The previously reported protonation of the dinuclear vanadium–dinitrogen complexes produced only ammonia, while **1**, **2**, **3**, and **4** produced both ammonia and hydrazine [21,22]. The yields of NH_3_ and N_2_H_4_ were estimated from the peak intensities of NH_4_^+^ using ^1^H NMR (7.04 ppm) and using the *p*-dimethylaminobenzaldehyde method using UV-vis spectra (458 nm), respectively (Appendix A). When using K[C_10_H_8_] as a reductant, the yields of NH_3_ and N_2_H_4_ were 47 and 11% for **1**, 38 and 16% for **2**, 78 and 7% for **3**, and 80 and 5% for **4** (per V atom), respectively (Table 3 and Appendix A). Their yields were higher than when using Na[C_10_H_8_]. Thus, the results suggest that the yield of products is dependent on the kind of alkali metal ions. The yield of NH_3_ for **2** was less than that of **1**, and the yield of N_2_H_4_ was higher than that of **1**. The result that the V–N_N2_ bond length is longer in **2** than in **1** suggests that mononuclear vanadium species are more likely to form from the vanadium–dinitrogen complex because the steric repulsion of substituents is greater in **2** than in **1**. Therefore, the intermediates of the protonation reaction were estimated to be a hetero-dinuclear Na^+^-/K^+^-V(μ-N_2_) complex, as in **4**. The yield with K[C_10_H_8_] as the reducing agent was higher than that with Na[C_10_H_8_], suggesting that potassium ions bound to N_2_ ligands are more readily exchanged to protons than sodium ions. We have recently found and reported similar behavior in the triamidoamine–chromium–dinitrogen system [49]. 

On the other hand, the yields of the protonation products of **3** and **4** were higher than those of **1** and **2**, and the yield of N_2_H_4_ was less than one tenth of that of NH_3_. This result suggests that protonation of mononuclear dinitrogen complexes gives higher yields of protonated products than dinuclear dinitrogen complexes and that the dicyclohexylmethyl groups in **3** or **4** surround the N_α_ atom of the N_2_ ligand, protecting it from proton attack and preventing the formation of N_2_H_4_ (Figure 5 right).

These findings suggest that the alkyl substituents on the secondary carbon atoms adjacent to the terminal N atom of triamidoamine stabilize the dimer structure, while the substituents on the tertiary carbon atoms destabilize the dimer structure or stabilize the monomer structure. These results also suggest that hetero-dinuclear dinitrogen complexes consisting of vanadium(II), alkali metal ions, and bridging N_2_ ligands are formed as intermediates in the protonation reactions of vanadium complexes **1**, **2**, and **3**. It was also found that hetero-dinuclear complexes produce both NH_3_ and N_2_H_4_, while the divanadium complexes produce only NH_3_. 

## 3. Materials and Methods

### 3.1. General Procedures

All manipulations were carried out under an inert N_2_ or Ar atmosphere using either a vacuum and N_2_/Ar gas manifold, or a MBraun MB 150B-G glovebox (N_2_/Ar). Reagents and solvents employed were commercially available. All anhydrous solvents were purchased from Wako Ltd. and were bubbled with argon to degas. The ligand tris(2-isopropylaminoethyl)amine (**H_3_L^iPr^**) was synthesized according to the literature methods [28].

### 3.2. Physical Measurements

^1^H-, ^13^C-, ^15^N-, and ^51^V-NMR spectra were recorded on a JEOL JNM-ECA500, or a JNM-ECA600 FT NMR spectrometer operating at 500 MHz (^1^H), at 125.77 MHz (^13^C) in C_6_D_6_ or DMSO-*d*_6_ at 298 K. ^1^H and ^13^C chemical shifts were referenced using residual protonated solvent resonance (C_6_D_6_: 7.16 ppm (^1^H) and 128.06 ppm (^13^C), DMSO-*d*_6_: 2.50 ppm (^1^H)). ^15^N and ^51^V NMR chemical shifts were externally referenced using HCONH_2_ (−266.712 ppm (^15^N)) and VOCl_3_ (0.00 ppm (^51^V)). Electronic absorption spectra were recorded on a JASCO V-770 spectrophotometer. FT-IR spectra were taken on an Agilent Cary 630 FTIR spectrophotometer. A resonance Raman spectroscopy was performed using a JASCO NRS-3300 spectrometer with 532 nm-wavelength Nd:YAG excitation source.

### 3.3. X-ray Crystallography Procedures

The data for **1**, **2**, and **3** were measured on Rigaku R-AXIS RAPID diffractometer using multi-layer mirror monochromated Mo *K*α (*λ* = 0.71073 Å) radiation. The data for **4** were measured on Rigaku R-AXIS RAPID II diffractometer using multi-layer mirror monochromated Cu *K*α (*λ* = 1.54178 Å) radiation. Crystal data and experimental details are listed in Appendix A. The calculations were performed with the Olex2 software package [50]. All structures were solved using ShelXT [51] structure solution program using the intrinsic phasing method, and the other atoms were found in subsequent Fourier maps. The structures were refined with ShelXL [52] using least squares minimization. All non-hydrogen atoms were anisotropically refined, unless otherwise stated. The hydrogen atoms were placed at their idealized positions, and the riding model was assumed, unless otherwise stated. CCDC-1970392 (**1**), 2167309 (**2**), 2167311 (**3**), 2167310 (**4**) contain the supplementary crystallographic data for this paper. These data can be obtained free of charge from The Cambridge Crystallographic Data Centre via www.ccdc.cam.ac.uk/data_request/cif, accessed on 22 August 2022.

### 3.4. Synthesis of Tris(2-(3-pentylamino)ethyl)amine (***H*_*3*_*L*^*Pen*^**)

Tris(2-aminoethyl)amine (14.6 g, 0.10 mol) was added to excess 3-pentanone (100 mL) and refluxed with a Dean–Stark trap overnight. The excess 3-pentanone was removed by evaporation. The mixture was cooled to 0 °C, and MeOH (50.0 mL) was added. Sodium borohydride (11.7 g, 0.31 mol) was added, and the mixture was stirred overnight at R.T. Sodium hydroxide (12.3 g, 0.31 mol) in water (100 mL) was added, and the aqueous layer was extracted 3 times by Et_2_O (100 mL). The organic layer was dried over Na_2_SO_4,_ and the solvent was removed by evaporation. The mixture was distilled to give a light-yellow oil (yield 33.1 g, 93%). ^1^H NMR (500 MHz, C_6_D_6_, 298 K): *δ* (ppm) 0.950 (t, 18H, –C*H*_3_), 1.45 (quin, 12H, CH–C*H*_2_–CH_3_), 2.37 (quin, 3H, C*H*–(CH_2_CH_3_)_2_), 2.50 (t, 6H, NH–CH_2_–C*H*_2_), 2.62 (t, 6H, NH–C*H*_2_–CH_2_). ^13^C{^1^H} NMR (125.77 MHz, C_6_D_6_, 298 K): *δ* (ppm) 10.18, 26.53, 45.34, 55.13, 60.64.

### 3.5. Synthesis of Tris(2-dicyclohexylmethylaminoethyl)amine (***H_3_L^Cy2^***)

Tris(2-aminoethyl)amine (14.6 g, 0.10 mol) was added to dicyclohexyl ketone (77.6 g, 0.40 mol) in toluene (100 mL) and refluxed with a Dean–Stark trap for 2 days. The excess dicyclohexyl ketone was removed in vacuo. The mixture was cooled to 0 °C, and MeOH (50.0 mL) was added. Sodium borohydride (11.7 g, 0.31 mol) was added, and the mixture was stirred overnight at R.T. Sodium hydroxide (12.3 g, 0.31 mol) in water (100 mL) was added, and the aqueous layer was extracted 3 times by Et_2_O (100 mL). The organic layer was dried over Na_2_SO_4_, and the solvent was removed by evaporation. The mixture MeOH (100 mL) and hexane (10 mL) was added and stored at −30 °C. The white solid was filtered and washed using solvent (2-propanol:hexane = 10:1). The white solid was resolved using CHCl_3_. Recrystallization was done by standing still in R.T. and white crystals were obtained (yield 43.5 g, 64%). ^1^H NMR (500 MHz, C_6_D_6_, 298 K): *δ* (ppm) 1.12–2.02 (m, 69H, *Cy*_2_–C*H*–), 2.62 (t, 6H, NH–CH_2_–C*H*_2_), 2.82 (t, 6H, NH–C*H*_2_–CH_2_). ^13^C{^1^H} NMR (125.77 MHz, C_6_D_6_, 298 K): *δ* (ppm) 27.23, 27.32, 27.38, 29.11, 31.81, 41.42, 50.68, 56.46, 68.90.

### 3.6. Synthesis of [{V(***L^iPr^***)}_2_(μ-^14^N_2_)] (***1***)

A 20 mL Schlenk flask was charged with **H_3_L^iPr^** (1.0 g, 3.7 mmol) and diethyl ether (4.00 mL) and cooled to −78 °C under nitrogen (^14^N_2_). *n*-butyllithium (4.2 mL, 11.0 mmol, 2.6 M in hexane) was added using a syringe. After 15 min, the reaction mixture was slowly warmed to 25 °C and was stirred for 1 h at room temperature. The reaction mixture was then cooled to −78 °C and VCl_3_THF_3_ (1.4 g, 3.7 mmol) was added via cannula to another Schlenk flask. The reaction mixture was again slowly warmed to 25 °C and was stirred overnight. The solvent was removed in vacuo, and the residue was extracted with diethyl ether (6 mL). The extract was filtered through Celite. The diethyl ether extract was transferred to a 20 mL Schlenk flask and stored in a fridge at −35 °C. [{VL^iPr^}_2_(*μ*-N_2_)] was obtained as dark green crystals (yield 0.57 g, 46%). ^1^H NMR (500 MHz, C_6_D_6_, 298 K): *δ* (ppm) 1.44 (d, 36H, –C*H*_3_), 2.40 (t, 12H, NH–CH_2_–C*H*_2_), 3.29 (t, 12H, N–C*H*_2_–CH_2_), 5.27 (sep, 6H, C*H*–(CH_3_)_2_), ^13^C{^1^H} NMR (125.77 MHz, C_6_D_6_, 298 K): *δ* (ppm) 23.46, 48.22, 53.11, 60.84, ^51^V NMR (131.56 MHz, C_6_D_6_, 298 K): *δ* (ppm) –211.1. FT IR (ATR, cm^–1^): 2957, 2916, 2892, 2851, 2788, 2652, 1457, 1438, 1380, 1364, 1347, 1332, 1328, 1282, 1259, 1239, 1202, 1155, 1129, 1099, 1077, 1056, 1028, 1002, 967, 939, 905, 862, 840, 818, 796, 789, 747, 618, 592, 572, 549, 495, 469. 

### 3.7. [{V(***L^iPr^***)}_2_(μ-^15^N_2_)] (***1**′***)

Complex **1′** was synthesized using the same method as complex **1** using ^15^N_2_ instead of ^14^N_2_. Complex **1′** was obtained as dark green crystals (yield 0.29 g, 24%). ^15^N-NMR (60.815 MHz, C_6_D_6_, 298 K): *δ* (ppm) 25.22. FT-IR (KBr, cm^–1^): 779 (*ν*(V–^15^N)).

### 3.8. Synthesis of [{V(***L^Pen^***)}_2_(μ-^14^N_2_)] (***2***)

Complex **2** was synthesized using the same method as complex **1** using **H_3_L^Pen^** instead of **H_3_L^iPr^**. Complex **2** was obtained as purple crystals (yield 1.1 g, 89%). ^1^H NMR (500 MHz, _C6_D_6_, 298 K): *δ* (ppm) 1.15 (t, 36H, –C*H*_3_), 1.74 (quin, 24H, CH–C*H*_2_–CH_3_), 2.37 (t, 12H, N–CH_2_–C*H*_2_), 3.20 (t, 6H, N–C*H*_2_–CH_2_), 4.77 (quin, 6H, C*H*–(CH_2_CH_3_)_2_). ^13^C{^1^H} NMR (125.77 MHz, C_6_D_6_, 298 K): *δ* (ppm) 12.51, 27.71, 48.57, 53.05, 71.52. ^51^V NMR (131.56 MHz, C_6_D_6_, 298 K): *δ* (ppm) –47.55. FT IR (ATR, cm^–1^): 2955, 2920, 2892, 2870, 2845, 2788, 1444, 1371, 1341, 1330, 1280, 1259, 1237, 1205, 1144, 1127, 1107, 1047, 1026, 993, 952, 905, 898, 866, 857, 834, 827, 812, 758, 728, 663, 642, 590, 559, 549, 572, 549, 527, 519, 486. 

### 3.9. Synthesis of [{V(***L^Pen^***)}_2_(μ-^15^N_2_)] (***2′***)

Complex **2′** was synthesized using the same method as complex **2** using ^15^N_2_ instead of ^14^N_2_. Complex **2′** was obtained as purple crystals (yield 0.49 g, 41%). ^15^N NMR (60.815 MHz, C_6_D_6_, 298 K): *δ* (ppm) 41.59. FT IR (ATR, cm^–1^): 715 (*ν*(V–^15^N)).

### 3.10. Synthesis of [V(***L^Cy2^***)] (***3***)

The complex **3** was synthesized using the same method as complex **1** using **H_3_L^Cy2^** in place of **H_3_L^iPr^**. THF for reaction solvent and hexane for recrystallization were used instead of diethyl ether (yield 0.79 g, 74%). FT IR (ATR, cm^–1^): 2955, 2920, 2892, 2870, 2845, 2788, 1444, 1371, 1341, 1330, 1280, 1259, 1237, 1205, 1144, 1127, 1107, 1047, 1026, 993, 952, 905, 898, 866, 857, 834, 827, 812, 758, 728, 663, 642, 590, 559, 549, 572, 549, 527, 519, 486.

### 3.11. Synthesis of [VK(***L^Cy2^***)(μ-^14^N_2_)(18-crown-6)] (***4***)

A 15 mL vial was inserted with complex **3** (0.20 g, 0.27 mmol) and THF (6 mL) under nitrogen gas. A quantity of K metals (0.11 g, 2.7 mmol, 10 eq.) was added to the solution, and the solution was stirred at R.T. overnight. The mixture was filtered through Celite, and the filtrate was added 18-crown-6-ether (79.8 mg, 0.30 mmol) in THF (1 mL). The mixture was stirred at R.T. for 1 min. Hexane (6 mL) was slowly added to the reaction mixture. Recrystallization of the complex **4** at −35 °C yielded a green crystal (0.13 mg, 39%). FT IR (ATR, cm^–1^): 1830, 1818 (*ν*(^14^N–^14^N)).

### 3.12. Synthesis of [VK(***L^Cy2^***)(μ-^15^N_2_)(18-crown-6)] (***4′***)

Complex **4′** was synthesized using the same method as complex **4** using ^15^N_2_ instead of ^14^N_2_. Complex **4′** was obtained as green crystals (60.2 mg, 18%). FT IR (ATR, cm^–1^): 1768, 1759 (*ν*(^15^N–^15^N)).

### 3.13. Protonation of ***1***, ***2***, ***3***, and ***4*** with Reductant and Proton Source

A 6 mL THF solution of the reductant (M[C_10_H_8_] (M = Na, K)), freshly prepared from alkali metal (1.1 mmol, 80 eq.) and naphthalene (0.15 mg, 1.1 mmol, 80 eq.) in a thick-walled glass bomb, was added to a 5 mL THF solution of **1** (10.0 mg, 1.5 × 10^−2^ mmol) at –78 °C, and the mixture was stirred for 1 h under N_2_. The reaction mixture turned from deep green to greenish brown. HOTf (0.17 g, 1.1 mmol) was added to the vigorously stirred reaction mixture of **1,** and the resultant solution was slowly warmed to 25 °C. After the solution was attired for 1 h at room temperature, the solvents were removed under reduced pressure to give a white solid containing ammonium and hydrazinium salts. The residue in the Schlenk tube was washed using diethyl ether and then, extracted using H_2_O (10 mL), and solvents of an aliquot (5 mL) of this solution were evaporated. The residue was analyzed using ^1^H NMR methods (for NH_3_). The other aliquot reacted with *p*-dimethylaminobenzaldehyde (for N_2_H_4_). 

### 3.14. NH_3_ Quantification Procedure

Quantification of ammonium salts was estimated using ^1^H NMR spectroscopy. The quantification of NH_4_^+^ was carried out using the method reported by Ashley and co-workers [53]. ^14^NH_4_^+^ was integrated relative to the vinylic protons of 2,5-dimethylfuran, contained within a DMSO-*d*_6_ capillary insert (*δ*:5.83, s, 2H), which was calibrated using a standard 5.2 × 10^−2^ M solution of NH_4_^+^ in DMSO-*d*_6_ (Appendix A and Appendix A).

### 3.15. N_2_H_4_ Quantification Procedure

The quantification of N_2_H_5_^+^ was carried out using the method reported by Ashley and co-workers [53]. The aliquots were analyzed for N_2_H_4_ via UV-vis spectroscopy using a standard spectrophotometric method, which reacted with an acidic *p*-dimethylaminobenzaldehyde solution and generated a yellow azine dye with a characteristic electronic absorption feature at 458 nm. The N_2_H_4_ in aliquot was quantified by comparison to the calibration curve (Appendix A).

## 4. Conclusions

In this study, we prepared vanadium(III) complexes with newly designed triamidoamine ligands ([L^R^]^3-^ R = iPr, Pen, and Cy_2_) (**1**, **2**, and **3**, respectively) under N_2_, which have tertiary C atoms adjacent to the terminal N atoms of the ligand. The X-ray structure analyses for **1**, **2**, and **3** revealed that complexes **1** and **2** were divanadium–dinitrogen complexes with a μ-N_2_ ligand between two vanadium ions in the end-on mode ([{V(L^R^)}_2_(N_2_)] R = iPr and Pen), while complex **3** was a mononuclear vanadium complex without N_2_ coordination, [V(L^Cy2^)]. These results indicate that complexes **1** and **2** with isopropyl and pentyl groups, respectively, can form dimeric structures, but due to steric repulsion between the dicyclohexylmethyl groups of the ligands, such a dimeric structure was not formed in complex **3**. The evidence of this is that all V–N_amido_ bond lengths around the vanadium(III) ion are the longest among complex **2** and the divanadium(III)–dinitrogen complexes with triamidoamine ligands reported so far. Furthermore, complex **3**, which has an even larger dicyclohexylmethyl group, did not form a dimer structure due to steric repulsion of the dicyclohexylmethyl group and formed a mononuclear complex. Reduction in complex **3** with potassium metal in the presence of 18-crown-6-ether produced the thermally unstable vanadium(II)–dinitrogen complex (**4**). The crystal structure of **4** was revealed to be a hetero-dinuclear complex with a bridging N_2_ ligand between V(II) and the K^+^ ion surrounded with 18-crown-6-ether.

Reactions of complexes **1**, **2**, **3**, and **4** with M[C_10_H_8_] (M = Na^+^ or K^+^) and HOTf yielded NH_3_ and N_2_H_4_, which are significantly different from our previous results using divanadium–dinitrogen complexes with similar triamidoamine ligands that we reported as releasing only ammonia [21,22]. This may be explained due to the following facts. In the case of the previously reported divanadium–dinitrogen complexes with a secondary C atom on the terminal N atom of the triamidoamine ligand, the addition of alkali metal ions resulted in the formation of a divanadium–dinitrogen complex as a reaction intermediate. On the other hand, in the cases of **1**, **2**, **3**, and **4** with tertiary C atoms instead of secondary C atoms, the addition of alkali metal ions resulted in the formation of a hetero-dinuclear nitrogen complex. Surrounding the N_α_ atom with large substituents increases the yield of NH_3_ and conversely, decreases the yield of N_2_H_4_. These results indicate that the large steric repulsion around the N_α_ atom prevents the protonation reaction into the N_α_ atom.

Here, we have found that changing the carbon atom adjacent to the terminal N atom of a triamidoamine ligand from a secondary carbon to a tertiary carbon can have a significant effect on structure and reactivity. These results provide an important finding to dinitrogen fixation catalysis and the model study of nitrogenase. 

## Data Availability

The data presented in this study are contained within this article and are supported by the data in the Appendix A.

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
