# Peer review of "The Steric Effect in Preparations of Vanadium(II)/(III) Dinitrogen Complexes of Triamidoamine Ligands Bearing Bulky Substituents"

_molecules, 2022, doi:10.3390/molecules27185864_

Round 1

Reviewer 1 Report

The paper of Yuji Kajita and co-authors is a fundamental work on synthesis new Vanadium(III) compounds with further activation of N2. Despite the fact that nitrogen activation by vanadium complexes is not uncommon, the article deserves publication in Molecules. The authors did a great job - three complexes with nitrogen were obtained. Their structure has been determined. I also liked the method of activation a nitrogen molecule through potassium and crown ether in the case of complex 3, where it was not possible to immediately fix nitrogen. The authors also protonated the obtained compounds, the reaction products were ammonia and hydrazine, the latter is rarer in such cases.

I have one general question - in the synthesis of the complex 3, THF was used instead of Et2O. Could the steric factor affect nitrogen fixation? How does this reaction proceed in Et2O?

The manuscript states that even at low temperatures in an inert atmospere, the complexes decompose, so are the resulting substances explosive when heated?

Line 238: NN2

Reviewer 2 Report

In this manuscript, Kokubo et al. present their findings in the field of mixed-valent complexes of vanadium bearing dinitrogen and other ligands. Apart of thorough characterization of all compounds using an extended set of modern phisical methods, the authors also examined the reactivity of these substances resulted in transformations of dinitrogen ligands. They provided sound discussion on the nature of differences in related reactivity.

I can see no flaws in terms of science. The manuscript could be accepted as it is since it is really interesting for specialists in advanced coordination chemistry.

My only criticism is related to language. I recommend seeking for assistance of someone with better language skills to re-read and correct the manuscript.
